# Pro-phagocytic function and structural basis of GPR84 signaling

Xuan Zhang[1,2,8], Yujing Wang [1,8], Shreyas Supekar [3,8], Xu Cao[4], Jingkai Zhou[4], Jessica Dang[4], Siqi Chen[4], Laura Jenkins[5], Sara Marsango[5], Xiu Li [1], Guibing Liu [1], Graeme Milligan [5] ✉, Mingye Feng [4] ✉, Hao Fan [3,6,7] ✉, Weimin Gong [1] ✉ & Cheng Zhang [2] ✉

GPR84 is a unique orphan G protein-coupled receptor (GPCR) that can be activated by endogenous medium-chain fatty acids (MCFAs). The signaling of GPR84 is largely pro-inflammatory, which can augment inflammatory response, and GPR84 also functions as a pro-phagocytic receptor to enhance phagocytic activities of macrophages. In this study, we show that the activation of GPR84 by the synthetic agonist 6-OAU can synergize with the blockade of CD47 on cancer cells to induce phagocytosis of cancer cells by macrophages. We also determine a high-resolution structure of the GPR84-$G_i$ signaling complex with 6-OAU. This structure reveals an occluded binding pocket for 6-OAU, the molecular basis of receptor activation involving non-conserved structural motifs of GPR84, and an unusual $G_i$-coupling interface. Together with computational docking and simulations studies, this structure also suggests a mechanism for the high selectivity of GPR84 for MCFAs and a potential routes of ligand binding and dissociation. These results provide a framework for understanding GPR84 signaling and developing new drugs targeting GPR84.

Free fatty acids (FFAs) are a unique group of lipid species, derived from triglycerides upon lipolysis. They can signal through a group of G protein-coupled receptors (GPCRs)[1,2] to function in metabolism, inflammation and immunity[3–6]. GPR84 is a $G_i$-coupled GPCR that has been suggested to recognize endogenous medium-chain fatty acids (MCFAs) but not short- or long-chain fatty acids (SCFAs and LCFAs)[7] (Supplementary Fig. 1a). Among native fatty acids, capric acid with a 10-carbon atom chain length showed the highest potency for activating GPR84[7]. Nevertheless, the low potency of those lipids and the lack of evidence suggesting the involvement of GPR84 in the physiological

function of MCFAs obscures their exclusive physiological pairing with the receptor[8]. Therefore, GPR84 still remains as an orphan GPCR. Nevertheless, GPR84 was found to be predominantly expressed by immune cells[7–9], and its expression can be strongly upregulated under inflammatory conditions to augment inflammatory responses and enhance phagocytosis[10–13]. Using synthetic GPR84 agonists and antagonists as useful pharmacological tools, previous research revealed the pro-inflammatory function of GPR84 signaling in various pathological conditions[11,13–15]. In particular, GPR84 signaling has been shown to promote fibrosis[15,16]. Several GPR84 antagonists were

[1]Division of Life Sciences and Medicine, University of Science and Technology of China, Hefei, Anhui, China. [2]Department of Pharmacology and Chemical Biology, University of Pittsburgh School of Medicine, University of Pittsburgh, Pittsburgh, PA 15261, USA. [3]Bioinformatics Institute (BII), Agency for Science, Technology and Research (A*STAR), Singapore 138671, Singapore. [4]Department of Immuno-Oncology, Beckman Research Institute, City of Hope Comprehensive Cancer Center, Duarte, CA 91010, USA. [5]Centre for Translational Pharmacology, School of Molecular Biosciences, College of Medical, Veterinary and Life Sciences, University of Glasgow, Glasgow G12 8QQ Scotland, UK. [6]Synthetic Biology Translational Research Program and Department of Biochemistry, School of Medicine, National University of Singapore, Singapore, Singapore. [7]Cancer and Stem Cell Biology Program, Duke-NUS Medical School, Singapore, Singapore. [8]These authors contributed equally: Xuan Zhang, Yujing Wang, Shreyas Supekar. ✉e-mail: Graeme.Milligan@glasgow.ac.uk; mfeng@coh.org; fanh@bii.a-star.edu.sg; wgong@ustc.edu.cn; chengzh@pitt.edu

developed for therapeutic purposes. Two of them, PBI-4050 and GLPG1205, have been tested in clinical trials for treating pulmonary fibrosis[17–20], although no significant therapeutic efficacy was reported so far.

One of the immunological functions of GPR84 signaling is to promote macrophage phagocytosis[10,21]. This has been indicated in a recent study for cancer cells[12]. This study identified an enzyme expressed in cancer cells named APMAP (Adipocyte Plasma Membrane Associated Protein) that functions as an anti-phagocytic factor to impede antibody-dependent cellular phagocytosis (ADCP) of cancer cells induced by blocking CD20[12]. Loss of the *APMAP* gene can significantly enhance the macrophage phagocytosis of cancer cells, which is dependent on GPR84 and $G_i$[12]. Analysis of previous RNA-sequencing data of human tumors also suggested specific expression of GPR84 in tumor-associated macrophages (TAMs)[12]. All the data suggested a critical role of the GPR84-$G_i$ signaling axis in mediating phagocytic activities of macrophages especially TAMs against cancer cells.

A major breakthrough in cancer immunosurveillance was the identification of 'don't eat me' signals such as CD47, which can be upregulated on cancer cells to inhibit macrophage phagocytosis[22,23]. Blocking the interaction between such signals and their macrophage-expressing receptors triggers cancer cell phagocytosis, leading to promising anticancer effects in mouse cancer models and clinical trials[22,23].

To further explore the therapeutic potential of activating GPR84 signaling in cancer, we first showed that activation of the GPR84-$G_i$ signaling axis by the commonly used synthetic GPR84 agonist 6-OAU (6-n-octylaminouracil)[7,11] could synergize with an anti-CD47 antibody[24,25] that disrupts the binding of CD47 to its receptor, Sirpα[23], on macrophages to induce phagocytosis of cancer cells by macrophages. To understand the actions of 6-OAU at a molecular level and to facilitate the potential rational development of other, more drug-like, GPR84 activators, we determined a high-resolution cryo-electron microscopy (cryo-EM) structure of the GPR84-$G_i$ signaling complex with 6-OAU. Our structure reveals a completely occluded binding pocket for 6-OAU and a receptor-specific Gi-coupling mode. Together with computational docking and simulations studies, this structure provides insights into lipid recognition by GPR84 and the receptor activation mechanism. We expect that our results will facilitate future drug development on GPR84 for cancer and other inflammatory diseases.

## Results

### Pro-phagocytic effect of GPR84-$G_i$ signaling in cancer cell phagocytosis by macrophages

Previous studies showed that GPR84 agonists could enhance the antibody-dependent cellular phagocytosis (ADCP) of B lymphocytes in the presence of an anti-CD20 antibody[12]. Here, we further tested the effect of 6-OAU with the CD47-blocking antibody, B6H12, in cancer cell phagocytosis by bone marrow-derived macrophages (BMDMs). Circulating monocytes that originate from bone marrow are constantly recruited to tumor sites and develop into TAMs. Therefore, BMDMs have been established as a sound model for studying phagocytosis of tumor cells. We used BMDMs from BALB/c mice whose Sirpα displays a binding affinity to human CD47 comparable to that of human Sirpα[26,27]. Our results indicated that treatment of BMDMs with 6-OAU promoted the phagocytosis of Raji cells, a human non-Hodgkin lymphoma cell line, in a concentration-dependent manner (Fig. 1a). To prove this effect was GPR84 dependent, we used the GPR84-specific antagonist GLPG1205[28] and showed that blocking GPR84 activation with GLPG1205 completely abolished the pro-phagocytic effect of 6-OAU (Fig. 1b). In addition, this effect of 6-OAU was also abolished by pre-treatment with the $G_i$ protein blocker pertussis toxin, confirming that the pro-phagocytic action of GPR84 is dependent on the $G_i$ signaling (Fig. 1b). We also knocked down the GPR84 expression in primary macrophages using CRISPR-Cas9 (Supplementary Fig. 1b and c) and showed that the pro-phagocytic effect of 6-OAU was abolished (Fig. 1c), further supporting that 6-OAU induces GPR84 signaling to promote phagocytosis. Altogether, our data suggested that activation of the GPR84-$G_i$ signaling axis in macrophages can synergize with CD47 blockade to drive the phagocytosis of cancer cells.

### Structure of the 6-OAU-GPR84-$G_i$ complex and an occluded ligand-binding pocket

To understand how 6-OAU activates the GPR84-$G_i$ signaling axis, we sought to determine a high-resolution structure of the 6-OAU-GPR84-$G_i$ complex by cryo-EM. We assembled the complex using the NanoBit tethering strategy in insect Sf9 cells[29]. The complex was treated with apyrase to hydrolyze GDP to ensure the α subunit of $G_i$, $G_{\alpha i}$, remained in a nucleotide-free state[30]. An antibody fragment, scFv16, was used to stabilize the $G_i$ heterotrimer[31]. The structure was determined to a global resolution of 3.0-Å by cryo-EM (Fig. 2, Supplementary Fig. 2, Supplementary Table. 1). The clear cryo-EM

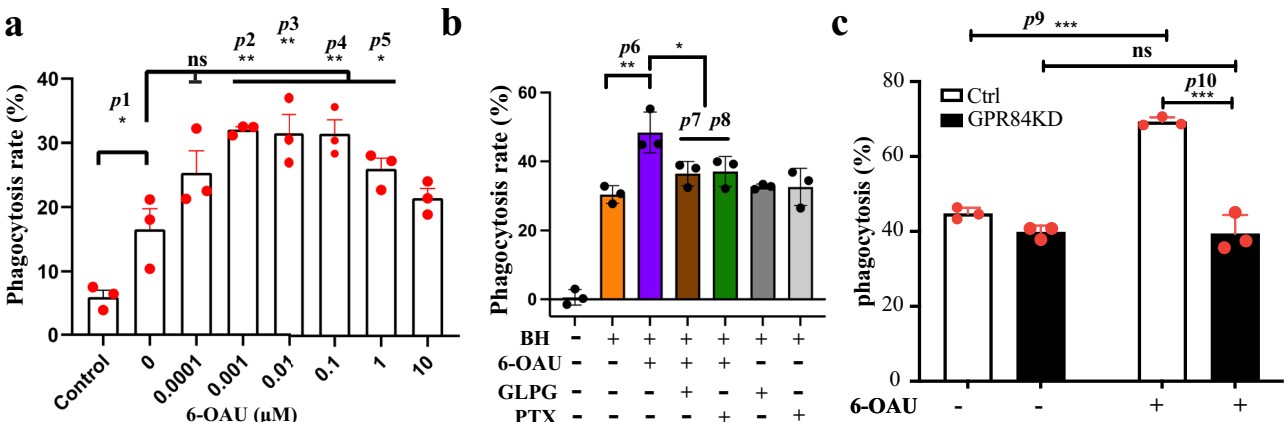

**Fig. 1 | GPR84-$G_i$ signaling facilitates cancer cell phagocytosis. a** Dose-dependent pro-phagocytic effect of 6-OAU. **b** GLPG1205 and pertussis toxin (PTX) abolished the pro-phagocytic effect of 6-OAU. BH means B6H12, the CD47-blocking antibody. PTX means pertussis toxin. **c** Knockdown of GPR84 expression in macrophages abolished the pro-phagocytic effect of 6-OAU. Each column represents means ± S.D. (*n* = 3) Data are representative of at least two independent experiments. Dunnett's one-way ANOVA test was performed to compare the means of two data groups. *$p < 0.1$, **$p < 0.01$, ***$p < 0.001$ (p1 = 0.0262, p2 = 0.0012, p3 = 0.0017, p4 = 0.0017, p5 = 0.0278, p6 = 0.0011, p7 = 0.0108, p8 = 0.0158, p9 = 0.000025, p10 = 0.000006). ns means no significant difference between the groups.

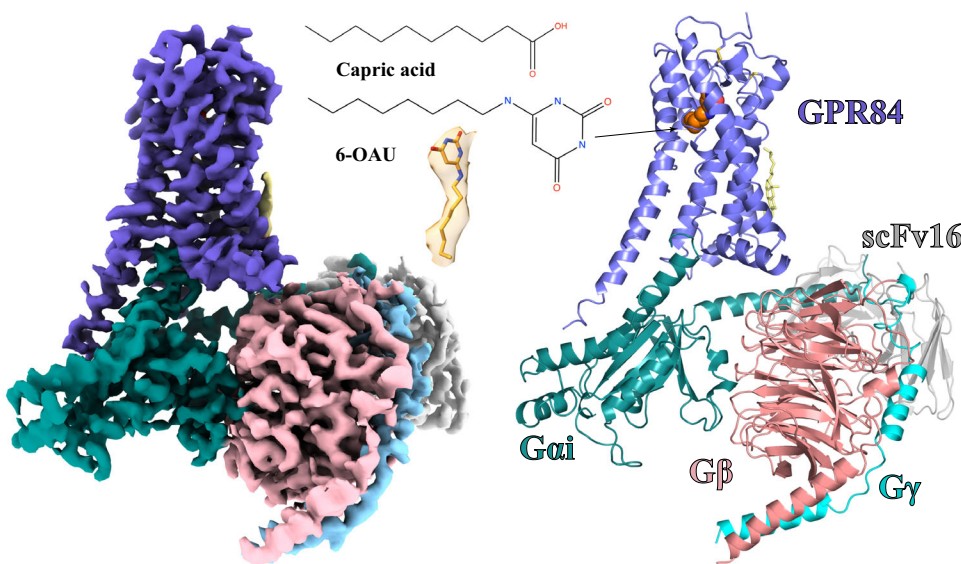

**Fig. 2 | Overall structure of the 6-OAU-GPR84-G$_i$ complex.** The left and right panels show the cryo-EM density map and the overall structure, respectively. The chemical structures of capric acid and 6-OAU and the cryo-EM density of 6-OAU contoured at xx are shown in the middle. GPR84 is colored in blue. Gαi, Gβ, and Gγ subunits are colored in cyan, pink, and light blue, respectively. ScFv16 is colored in gray.

density of the receptor allowed us to model the residues from D6 to P389 of GPR84 except for the long intracellular loop 3 (ICL3) from L217 to F314 in the structure. For the heterotrimeric G$_i$ protein, the helical domain of G$_{αi}$ was not modeled due to potential structural flexibility[32].

The overall structure of GPR84 resembles those of other Class A rhodopsin-like GPCRs[33]. The extracellular loop 2 (ECL2), which is almost perpendicular to the 7-transmembrane helical bundle (7-TM), adopts a β-hairpin structure to extend towards transmembrane helix 1 (TM1) on top of the 6-OAU binding pocket, shielding it from the extracellular milieu (Fig. 3a). Two disulfide bonds further stabilize the conformation of ECL2; One forms between C168 of ECL2 and C93[3.25] (superscripts represent Ballesteros-Weinstein numbering[34]) of TM3, which is highly conserved in Class A GPCRs[35], and the other forms between C166 of ECL2 and the N-terminal residue C11. The latter has also been proposed in a previous modeling study[36]. The importance of the highly conserved disulfide bridge in class A GPCR cell surface delivery and function is well established. To assess the contribution of the additional disulfide bond between C11 and C166, we mutated residue C11 to Ala. No response to 6-OAU was observed when this mutant was expressed transiently in HEK293 cells (Fig. 3b). The basis for lack of function of 6-OAU at this mutant however remains uncertain. Compound 38 (9-(2-phenylethyl)−2-(2-pyrazin-2-yloxyethoxy)−6,7-dihydropyrimido [6,1-a]isoquinolin-4-one)[28] is an allosteric antagonist of GPR84, closely related to GLPG1205, with nanomolar affinity. Despite [³H]38 displaying high affinity specific binding to wild-type GPR84, in parallel experiments no specific binding of this radioligand to GPR84 C11A mutant was observed (Fig. S3a). To complement these studies, we also performed specific binding studies using the GPR84 orthosteric antagonist [³H]3-((5,6-diphenyl-1, 2,4-triazin-3-yl)methyl)−1*H*-indole ([³H]140)[37]. This radioligand also lacked high affinity binding at GPR84 C11A GPR84 (Supplementary Fig. 3a). It is thus likely that without this disulfide bridge GPR84 fails to fold correctly. It is also the case for another residue H352[7.35], which is located near the 6-OAU binding pocket but doesn't interact with 6-OAU (Supplementary Fig. 3b). We initially expected that mutations of this residue wouldn't disrupt 6-OAU function since it is not involved in ligand binding. However, it turned out that the H352A mutant didn't exhibit any specific binding of both radioligands (Supplementary Fig. 3a), indicating potential protein misfolding.

No openings between transmembrane helices are observed around 6-OAU. As a result, the ligand is completely buried inside the 7-TM and occluded from the outside aqueous and lipidic environment (Fig. 3a and Supplementary Fig. 3b). A similar completely occluded ligand-binding pocket has also been observed in another lipid GPCR, the cannabinoid receptor 2 (CB2)[38,39] (Supplementary Fig. 3b, c). However, different from GPR84, in the structure of active CB2 with G$_i$, a part of the N-terminal region of CB2 folds on top of the ligand-binding pocket to shield it from the extracellular environment[38,39] (Supplementary Fig. 3c).

6-OAU is an amphipathic molecule with a polar head group and an octylamine tail (Fig. 1). Accordingly, multiple polar and hydrophobic interactions between 6-OAU and GPR84 are observed (Fig. 3c, Supplementary Fig. 4). The uracil head group of 6-OAU engage in extensive hydrogen-bonding interactions with T167, S169 and R172 in ECL2 and Y69[2.53] and W360[7.43] of GPR84. The amine group of the octylamine tail of 6-OAU also forms a salt bridge with N104[3.36]. The mutation of T167A has been shown to abolish the action of capric acid[40]. We also found that mutations of S169A, W360A, and R172A could make the receptor much less responsive to 6-OAU (Fig. 3b), proving the important roles of the polar interactions with GPR84 in the agonistic action of 6-OAU. We again probed the expression of these mutants by assessing specific binding of [³H]38 and [³H]140. Our results revealed that the expression of R172A was akin to wild-type whilst detection of the expression of W360A was negligible (Supplementary Fig. 3a). Previous modeling studies on the binding mode of [³H]140 and other related 1,2,4 triazine GPR84 orthosteric antagonists have suggested an important role for W360[7.43], which was supported by the loss of ligand binding at W360A[36]. Our studies suggest a similarly important role of W360[7.43] in the recognition of [³H]38 and potentially other GLPG1205-related non-competitive antagonists. Interestingly, R172K, which was also effectively detected by binding of [³H]38 and [³H]140 (Supplementary Fig. 3a), caused as extensive a change of the EC$_{50}$ of 6-OAU as that caused by R172A (Fig. 3b). It is possible that R172K may result in new interactions and cause conformational changes of ECL2 to disrupt 6-OAU binding.

In addition to the polar interactions, the saturated octyl tail of 6-OAU resides in a hydrophobic sub-pocket surrounded by GPR84 residues F101[3.33], F152[4.57], L182[5.42], Y186[5.46], Y332[6.48], F335[6.51], L336[6.52], and L361[7.44] (Fig. 3c, Supplementary Fig. 4). Consistent with such finding,

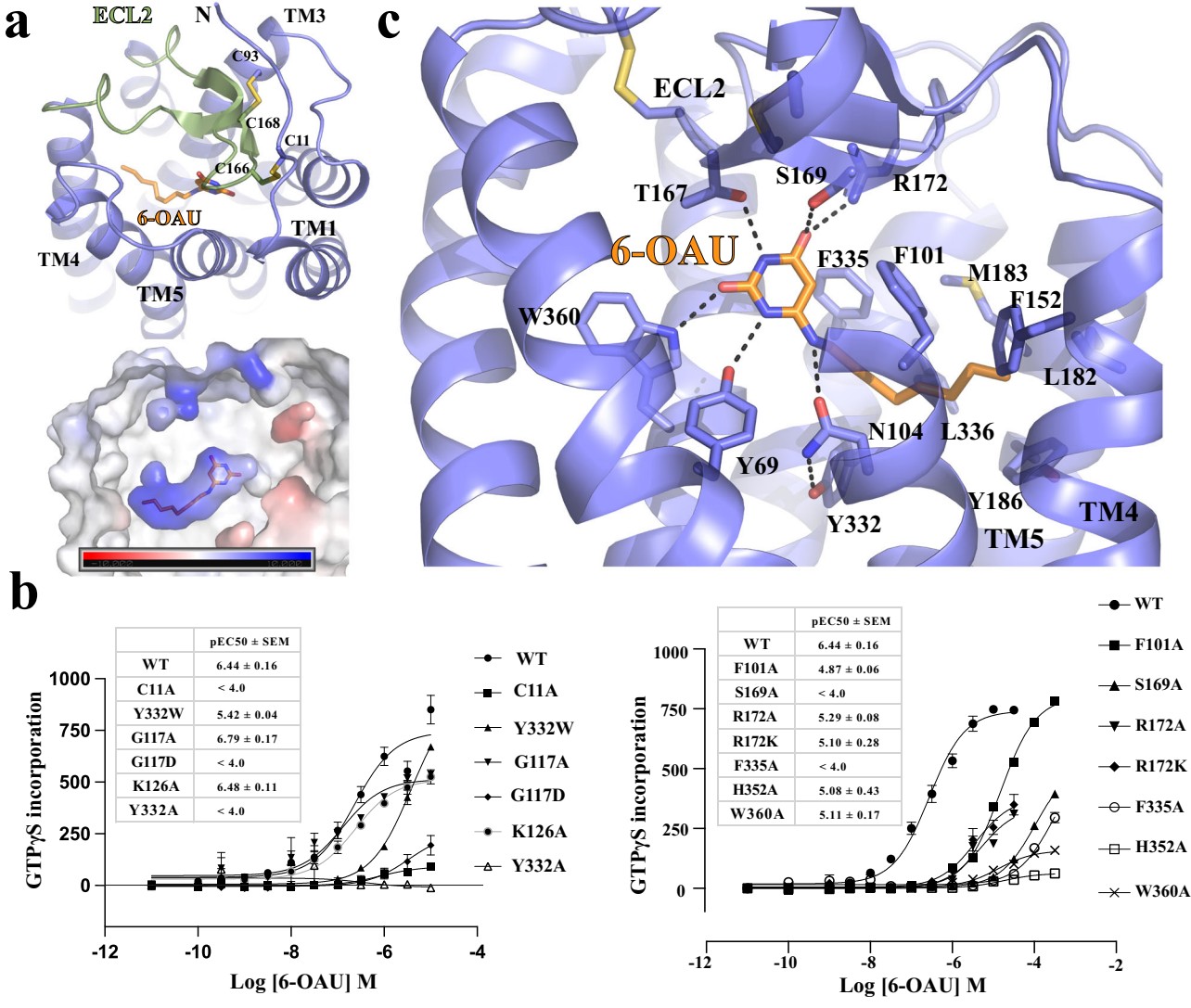

**Fig. 3 | 6-OAU binding in GPR84. a** Occluded binding pocket for 6-OAU covered by ECL2. The lower panel indicates the charge potential of the 6-OAU binding pocket. The bar shows the levels of negative (red) and positive (blue) charge potential. **b** Mutagenesis data using GTPγS incorporation assays. Data represent mean ± S.E.M. from at least three independent experiments. **c** Interactions between 6-OAU and GPR84. The polar interactions are shown as dashed lines. 6-OAU is shown as orange sticks in all figures.

our mutagenesis studies showed that F101A and F335A resulted in much-compromised action of 6-OAU (Fig. 3b). We have previously shown that both F101[3.33] and F335[6.51] play important roles in binding of the orthosteric 1,2,4 triazine antagonists[36], here further illustrated by the lack of specific binding of [³H]140 to F335A (Supplementary Fig. 3a). Of great interest, however, binding of the non-competitive antagonist [³H]38 was unaffected by this mutation (Supplementary Fig. 3a). Although direct observation of the binding modes of the two antagonist classes is lacking, these studies confirm, as anticipated from their non-competitive versus competitive actions, that they clearly differ.

The overall binding pose of 6-OAU is similar to those of leukotriene B4 (LTB4)[41], sphingosine 1-phosphate (S1P)[42-45], lysophosphatidic acid (LPA)[46], and prostaglandin E2 (PGE2)[47] in their respective GPCRs (Fig. 4). In the structures of these four lipids with their receptors, the carboxylate head group of each lipid is located near the extracellular surface while the hydrophobic carbon chains are buried inside the 7-TM bundle (Fig. 4). The binding pockets of all four lipids have openings at the extracellular regions of their respective receptors, potentially serving as the ligand entrance (Fig. 4). This is in contrast to the occluded binding pocket of 6-OAU. Also, in GPR84, ECL2 inserts

into the 7-TM region, resulting in a much shorter binding pocket compared to those in the receptors for LTB4, S1P, LPA, and PGE2 (Fig. 4), explaining why GPR84 doesn't bind to LCFAs[7]. Analysis of the charge potential of the 6-OAU binding pocket showed an uneven positive charge distribution (Fig. 3b). A similar uneven distribution of the positive charge potential was observed for the ligand-binding pocket in the prostaglandin D2 (PGD2) receptor DP2, which has been proposed to facilitate the recognition of PGD2 by DP2[48,49]. For GPR84, previous studies suggested that the positive charge of R172 in the ECL2 plays a key role in the binding of MCFAs by coordinating the carboxylate head group[50,51].

## Ligand recognition mechanisms revealed by computational docking and MD simulations

To further investigate how GPR84 recognizes different agonists, we sought to dock three other GPR84 agonists, embelin, capric acid, and 2-hydroxy capric acid, to the GPR84 structure. To validate our docking methods, we first docked 6-OAU to our structure, which recapitulated the 6-OAU binding pose observed in our structure with slight differences at the lipid tail, implying a high flexibility of this part (Supplementary Fig. 5a). Our docking results showed that embelin, capric acid,

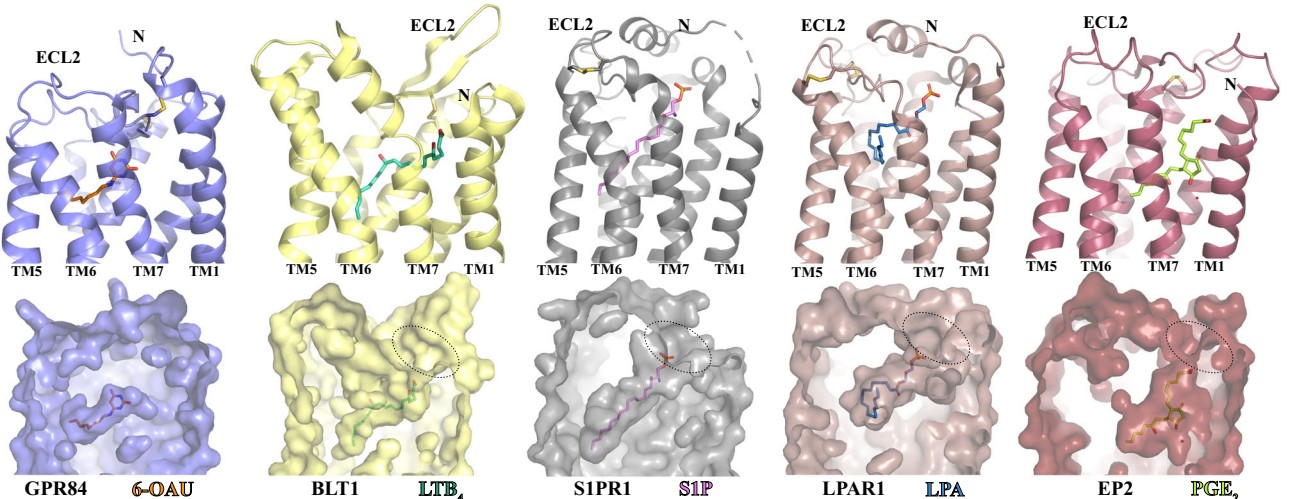

**Fig. 4 | Comparison of the ligand-binding pockets in GPR84 and four other lipid GPCRs.** BLT1, S1PR1, LPAR1, and EP2 are receptors of LTB₄, S1P, LPA, and PGE₂, respectively. The structures of GPR84, BLT1 (PDB ID 7VKT), S1PR1 (PDB ID 7TD3), LPAR1 (PDB ID 7TD0), and EP2 (PDB ID 7CX2) are colored slate, light yellow, gray, brown, and dark red, respectively. All ligands are shown in sticks. In each column, the cartoon model and surface representation are used for the same receptor. The structures of the five receptors are placed side by side in each column after structural alignment, providing a consistent viewpoint from the same angle.

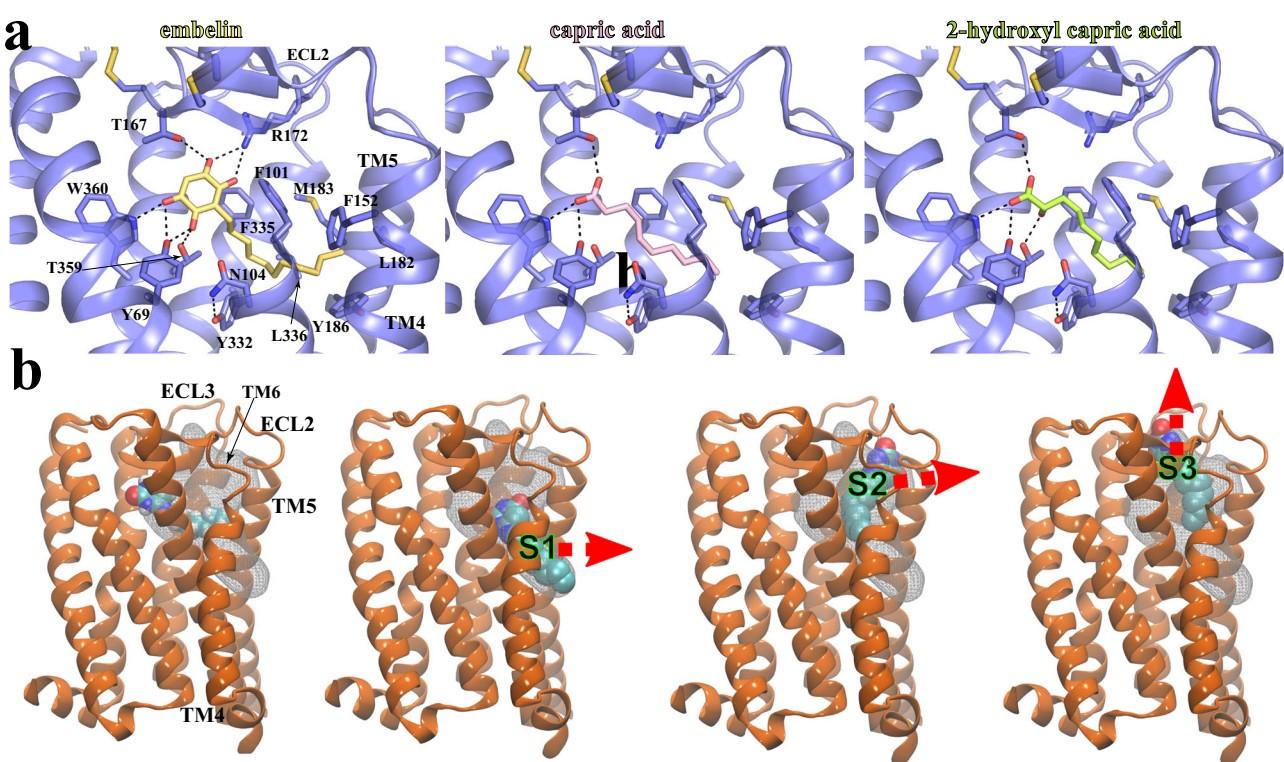

**Fig. 5 | Docking of GPR84 agonists and MD simulations of 6-OAU-bound GPR84.** **a** Interactions of docked embelin (light yellow), capric acid (pink), and 2-hydroxyl capric acid (lime) with GPR84 (slate). Hydrogen bonds are shown as black dashed lines. **b** Putative exit routes of 6-OAU in MD simulations. The 6-OAU movement during the simulations is shown as density in white grid. Red arrows indicate possible ligand exit routes via metastable sites S1, S2 or S3. 6-OAU is shown as cyan spheres and GPR84 is shown in orange.

and 2-hydroxy capric acid adopt similar binding poses as 6-OAU (Supplementary Fig. 5b), in which their polar groups located near ECL2 engage in different sets of hydrogen-bonding interactions with nearby GPR84 residues and their lipid tails stick into the same hydrophobic pocket towards the cleft between TM4 and TM5 (Fig. 5a). GPR84 residues T167 in ECL2, Y69[2.53], and W360[7.43] are involved in the hydrogen boding interactions with all four agonists (Figs. 3c and 5a).

The docking scores for these four agonists (Supplementary Table. 2) suggest the ranking of their affinities as the following: 6-OAU > embelin > capric acid ≈ 2-hydroxy capric acid, which is in line with their reported EC₅₀ values in the literature[8,52].

To investigate the ligand-binding process, we performed large-scale (*ca.* 20 μs) molecular dynamics (MD) simulations of GPR84 in apo (GPR84 alone) and holo (GPR84 with 6-OAU) states. In the holo state

simulations, we observed that 6-OAU primarily occupies the native binding pocket (Fig. 5b). However, we found that in several instances 6-OAU indeed moved away from the native state to occupy other metastable sites on the periphery of GPR84 (Fig. 5b, Supplementary Fig. 6a). The first metastable site, namely, site 1 (S1), was located at the interface among TM4-TM5 and membrane lipids (Fig. 5b), where 6-OAU made hydrophobic contacts with membrane lipids and GPR84 residues (Supplementary Fig. 6b). The second metastable site, namely, site 2 (S2), was located at the interface at the TM5-TM6 interface (Fig. 5b), where 6-OAU made H-bonds with membrane lipid head-groups and hydrophobic contacts with GPR84 (Supplementary Fig. 6c). The third metastable site, namely, site 3 (S3), was located on top of the orthosteric site near ECL2-ECL3-water interface (Fig. 5b). At site 3, R172 at the base of ECL2 β-hairpin made a cation-π interaction with 6-OAU, presumably acting as a gatekeeper residue preventing 6-OAU to escape to the solution phase (Supplementary Fig. 6d). The identified peripheral sites suggested putative routes for 6-OAU to exit from the orthosteric site via sites 1 or 2 to the membrane phase, or via site 3 to the extracellular milieu (Fig. 5b, Supplementary Fig. 6a)[53]. We also performed holo MD simulations of GPR84 with embelin, capric acid and 2-hydroxy capric acids, in the same time-scale as 6-OAU. Similarly, we found that besides primarily occupying the native binding pocket, all the ligands also transiently explored the metastable states S1, S2 and S3 as those observed in the 6-OAU holo simulations (Supplementary Fig. 7). This is supported by the distributions of distance between center-of-mass (COM) of the native pocket and the COM of the respective ligands (Supplementary Fig. 8a) with majority of the density within 5 Å representing the native binding pocket but also a minority sampling in the long tail region representing metastable states as described above. It should be noted that a direct comparison is not possible using this metric as the four ligands have different COMs owing to different headgroups. For a direct comparison of ligand dynamics, we also calculated root-mean-squared-deviation (RMSD) of each ligand during MD simulations using the respective crystal structure or docking pose as the reference (Supplementary Fig. 8b). The distributions of RMSD also have peaks within 5 Å and long tails. Interestingly, similar to the docking predicted affinities above, the COM distances and the RMSD plots also suggest affinities: 6-OAU > embelin > capric acid ≈ 2-hydroxy capric acid, which is in line with their reported $EC_{50}$ values in the literature[8,52].

## Non-conserved structural motifs of GPR84 and receptor activation

Since there is no experimentally solved inactive structure of GPR84, we used the Alphafold predicted structure of apo GPR84[54,55] in our structural comparison analysis. This structure is expected to represent an inactive conformation since there is no agonist or G protein in the structure. Indeed, structural alignment indicated large conformational rearrangements at the cytoplasmic region including a large outward displacement of TM6 and an inward movement of TM7 of the active GPR84 compared to the Alphafold predicted structure (Fig. 6a). These features are characteristic of receptor activation for Class A GPCRs[56]. In contrast, the extracellular region of GPR84 only showed subtle differences between these two structures (Fig. 6a). It is to be noted that Alphafold successfully predicted the unusual conformation of ECL2 of GPR84 (Fig. 6a)[36].

For Class A GPCRs, conserved residues $W^{6.48}$ and $F^{6.44}$ form a 'transmission switch' motif that connects the extracellular agonist-binding events to the conformational changes at the cytoplasmic regions during receptor activation[57,58]. In GPR84, while $F^{6.44}$ is conserved, $W^{6.48}$ is replaced by a tyrosine residue, $Y332^{6.48}$, which forms a hydrogen bond with $N104^{3.36}$ (Fig. 6b). In the Alphafold predicted structure, $Y332^{6.48}$ also forms hydrogen bonds with $N104^{3.36}$ (Fig. 6b). Structural alignment with the active GPR84 structure showed large rearrangements of these two residues due to the steric effects caused by the octyl tail of 6-OAU (Fig. 6b). It is likely that 6-OAU activates GPR84 mainly by inducing conformational changes of the $Y332^{6.48}$-$N104^{3.36}$ pair, which in turn induce significant displacements of $F328^{6.44}$ and the cytoplasmic segment of TM6 (Fig. 6b). The conformational change of $Y332^{6.48}$ also causes the swing of the side chain of the TM7 residue $N362^{7.45}$. This further results in the formation of a hydrogen-bonding network mediated by $N362^{7.45}$ and surrounding residues $S107^{3.39}$, $Y332^{6.48}$, and $N366^{7.49}$ (Fig. 6b), potentially leading to the inward movement of TM7 for $G_i$-coupling (Fig. 6b). Such a network is missing in the Alphafold predicted structure (Supplementary Fig. 9a). Alteration of $Y332^{6.48}$ to Ala resulted in a lack of response to the agonist (Fig. 3b). In contrast, alteration of $Y332^{6.48}$ to the more commonly found Trp reduced potency by more than 10 fold but did not ablate the function of 6-OAU (Fig. 3b). In addition, $N366^{7.49}$ is a part of the conserved $N^{7.49}P^{7.50}$xxY motif[58-60]. This residue forms a salt bridge with $D66^{2.50}$ in the Alphafold predicted inactive structure (Supplementary Fig. 9a). Both residues have been shown to coordinate with a sodium

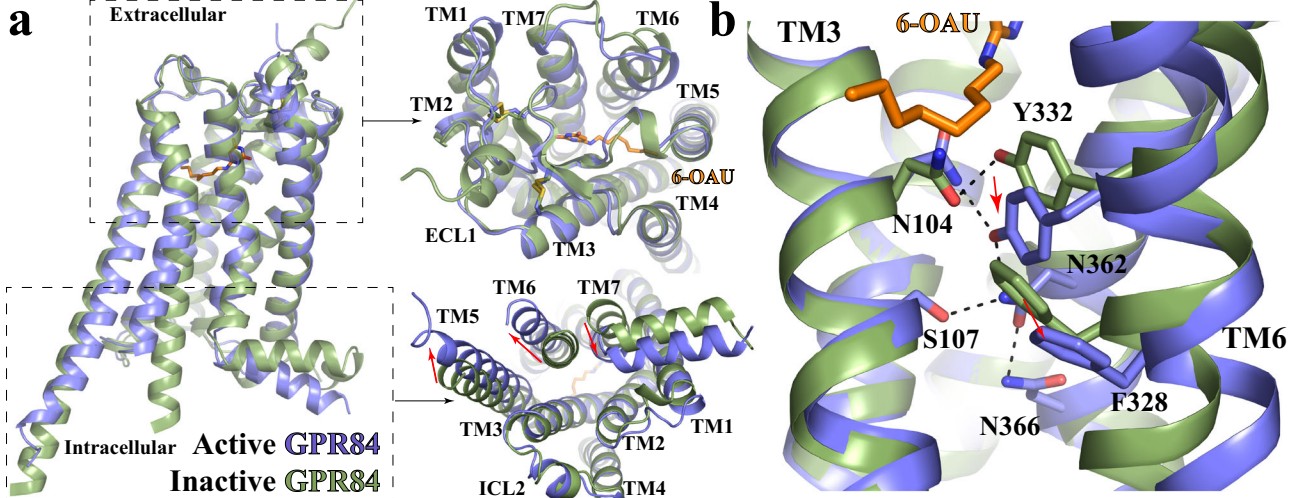

**Fig. 6 | Active conformation of GPR84. a** Superimposition of the active GPR84 structure (blue) to the Alphafold predicted inactive GPR84 structure (green). The extracellular and intracellular regions are shown in the left upper and lower panels, respectively. The red arrows indicate conformational changes of TMs. **b** Residues involved in the receptor activation at the core region of GPR84.

ion in the inactive structures of many other Class A GPCRs, and collapse of this sodium coordination site is involved in the receptor activation[61,62]. Indeed, in the active structure of GPR84, N366[7.49] moves away from D66[2.50], which may result from the conformational changes of TM7 in receptor activation.

Another highly conserved structural motif of Class A GPCRs that is not conserved in GPR84 is the D/E[3.49]R[3.50]Y motif. This motif is located near the cytoplasmic surface that mediates intrahelical interactions believed to stabilize the inactive conformation of receptors or modulate receptor activation and G protein coupling[63,64], which is replaced by G117[3.49]R[3.50]Y in GPR84 (Supplementary Fig. 9b). Two phenylalanine residues F128 and F132 in the intracellular loop 2 (ICL2) and F55[2.39] in TM2 are in the close vicinity of G117[3.49] (Supplementary Fig. 9b). They would cause steric clashes if G117[3.49] is replaced by a glutamic (E) or aspartic (D) acid residue. Indeed, alteration of G117 to D eliminated function of 6-OAU whereas alteration of G117 to A didn't significantly affect 6-OAU signaling (Fig. 3b). For both G117A and G117D specific binding of both the allosteric antagonist [³H]38 and the orthosteric antagonist [³H]140 confirmed successful expression of these mutants (Supplementary Fig. 3a). Interestingly, F128 and F132 in ICL2 form a hydrophobic cluster with F55[2.39] in TM2 and L121[3.53] in TM3, potentially stabilizing the α-helical structure of ICL2 (Supplementary Fig. 9b). Such a helical structure of ICL2 is also present in the Alphafold predicted inactive structure of GPR84 (Fig. 6a). This is in contrast to the loop structure of ICL2 in many other Class A GPCRs in the inactive conformation[65].

## Gᵢ coupling mode

In the structure of 6-OAU-bound GPR84-Gᵢ complex, Gᵢ couples to GPR84 in a canonical way similar to that in the structures of other Gᵢ-coupled GPCRs. The C-terminal α-helix, α5, of Gαᵢ is the major interaction site for GPR84 (Supplementary Fig. 10a). In the C-terminal half of α5 of Gαᵢ, residues I344, L348, and L353 in the α5 and the last residue, F354, of Gαᵢ form hydrophobic interactions with I122[3.54], I201[5.62], V205[5.66], and V317[6.33] of GPR84 (Fig. 7a). R118[3.50] of GPR84 in the non-conserved GR[3.50]Y motif mediates a hydrogen-bonding interaction network by interacting with Y198[5.59] and Y370[7.53] of GPR84 and with the main-chain carbonyl of C351 of Gαᵢ, while Q376 in the helix 8 of GPR84 forms hydrogen bonds with the main-chain carbonyl of K349 and the side chain of D350 of Gᵢ (Fig. 7a). The Gβ subunit of Gᵢ is also involved in

direct interactions with GPR84. D312 of Gβ forms salt bridges with K50 and R387 from ICL1 and helix 8, respectively, of GPR84, and K386 from helix 8 of GPR84 forms a cation-π interaction with F292 of Gβ (Fig. 7b).

There are some unique features of interactions with Gᵢ observed for GPR84. First, the TM5 is much longer than any of the other TMs of GPR84 (Fig. 2). As a result, Y215[5.76] at the C-terminal end of TM5 of GPR84 forms aromatic and polar interactions with residues F334 and D337 in the C-terminal half of α5 of Gαᵢ, respectively (Fig. 7c). Another residue, H322, in the β-strand β6 of Gαᵢ is also involved in π-π interactions with Y215[5.76] of GPR84 (Fig. 7c). All of those interactions may facilitate the displacement of α5 of Gαᵢ, which is translated to the conformational changes of the β6-α5 loop and the release of GDP in Gᵢ activation[66] (Supplementary Fig. 10b). Second, in most of other Gᵢ-coupled GPCR structures, the position 34.51 in the ICL2 is usually a hydrophobic residue that forms hydrophobic interactions with residues including L194 and I343 in Gαᵢ (Supplementary Fig. 10c). In GPR84, this position is K126 (Supplementary Fig. S10c). As a result, there are no direct interactions between ICL2 of GPR84 and Gαᵢ. Nevertheless, the alteration of this residue to Ala did not affect the potency and function of 6-OAU (Fig. 3b) or the expression of the modified receptor (Supplementary Fig. 3a).

## Discussion

Our results offer insight into the ligand recognition mechanism for GPR84. First, in our structure, the conformation of ECL2 results in a ligand-binding pocket with a size that cannot accommodate LCFAs with 14 or more carbons. In addition, for a potential fatty acid agonist of GPR84, the lipid moiety needs to reach the bottom region of the binding pocket in order to cause conformational changes of residues including Y332[6.48] at the core region to activate the receptor. Therefore, the unique shape and size of the binding pocket of GPR84 well explain the preference of the receptor for MCFAs over LCFAs or SCFAs. Second, the occluded binding pocket for 6-OAU makes it difficult to propose a ligand entrance in GPR84. Our MD simulations results suggested three possible routes for 6-OAU to exit the receptor, all of which require conformational changes of the 7-TM region or the extracellular loop region. Interestingly, in the Alphafold predicted inactive structure of GPR84, there are small openings at the extracellular surface between ECL2 and ECL3 and the helical surface between TM5 and TM6 (Supplementary Fig. 11), resembling the S3 and

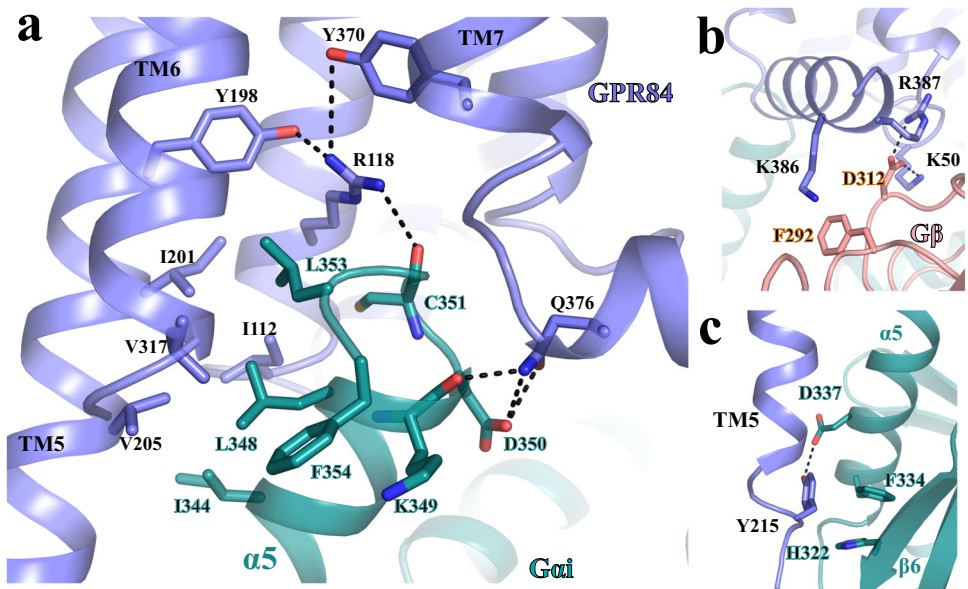

**Fig. 7 | Gᵢ-coupling to GPR84. a** Interactions between GPR84 (blue) and the α5 of Gαᵢ (cyan). **b** Interactions between GPR84 (blue) and Gβ (salmon). **c** Interactions between the C-terminal end of TM5 of GPR84 (blue) and Gαᵢ (cyan). All polar interactions are shown as dashed lines.

S2 metastable sites in our MD simulations (Fig. 5b and Supplementary Figs. 6, 7). It is likely that the extracellular region of TM5 or TM6 undergoes conformational changes to result in the S2 or S3 site serving as the ligand entrance for the endogenous and synthetic GPR84 ligands.

Tissue macrophages use multiple phagocytic receptors including several opsonic receptors, pattern-recognition receptors (PRRs), and receptor tyrosine kinases (RTKs) to initiate the process of phagocytosis against pathogens (foreign) and apoptotic cells (self)[67,68]. Previous studies[10,12] and ours suggested that GPR84 serves as a new phagocytic receptor in inflammatory conditions. In particular, the ability of the GPR84 agonist 6-OAU to promote phagocytosis of cancer cells induced by CD47 blockage and the specific expression of GPR84 in TAMs[12] implied a potential role of GPR84 in cancer immune surveillance. Furthermore, we demonstrated that the $G_i$ signaling pathway is critical in the phagocytic function of GPR84 against cancer cells. $G_i$ pathway-selective GPR84 agonists, or $G_i$-biased GPR84 agonists, may offer an interesting therapeutic method to enhance the phagocytosis of cancer cells by macrophages. It has been shown that GPR84 agonists such as 6-OAU could effectively recruit β-arrestins[21,51], the classic scaffold proteins promoting GPCR internalization and desensitization[69]. Indeed, in our assays, high concentration of 6-OAU led to lowered levels of phagocytosis (Fig. 1a), which was likely due to GPR84 desensitization[70]. In this regard, selective activation of the GPR84-$G_i$ pathway with minimal β-arrestin recruitment by $G_i$-biased GPR84 agonists such as DL-175[21] or PSB-16671[51] may promote more sustained macrophage phagocytosis of cancer cells compared to 6-OAU. In addition, the membrane-embedded enzyme APMAP that is highly expressed on the surface of cancer cells has been proposed to degrade the physiological lipid ligand of GPR84 to negatively regulate macrophage phagocytosis[12]. Identifying such a ligand of GPR84 and APMAP may lead to the identification of a novel pathway regulating macrophage function and facilitate the development of novel therapeutics targeting this pathway in addition to GPR84 activators to enhance cancer cell phagocytosis.

## Methods

### Macrophage phagocytosis assay

The phagocytic ability of macrophages toward live cancer cells was evaluated by a luminescence-based long-term phagocytosis assay as we previously described[71]. Specifically, luciferase-expressing Raji cells (ATCC, Cat# CCL-86) were co-cultured with BMDMs isolated from mouse blood for 24 h in the absence or presence of CD47-blocking antibody (clone B6H12) (BioXCell, Cat# BE0019-1). Thereafter, the luminescence signal was measured by the addition of luciferin and detection with Cytation 3. For evaluating the effects of GPR84 agonists and/or antagonists, BMDMs were pretreated with corresponding chemicals overnight. After a thorough wash with PBS, the pretreated BMDMs were then used for phagocytosis assay. Cancer cells cultured without BMDMs were used as a normalization control for calculation which indicates a phagocytosis rate of 0%. 6-OAU (0.1 μM) was used to stimulate the activity of GPR84, while GLPG1205 (10 μM) or pertussis toxin (0.1 mg/ml) were used to block the stimulative effect of 6-OUA. Mycoplasma examination was performed routinely for Raji cells and the result was negative.

The CRISPR-Cas9 system was used to knockdown the GPR84 expression in primary macrophages. The control sgRNA (AGUCCG-GUCGAAAUCUGUAU), sgRNA targeting mouse GPR84 (CGCCA-GUUUCGCCACGCGUA) were cloned into LentiCRISPR V2 vector. The plasmids were transfected with the packaging and envelop plasmids into 293 T cells to generate lentiviruses. Forty-eight hours transfection, the viruses were harvested and incubated with 4x Lentivirus Concentrator Solution containing 40% PEG-8000 and 1.2 M NaCl with constant rocking overnight at 4 °C. After incubation, the virus was centrifuged at 1600×g for 60 min at 4 °C, and thoroughly resuspended with Iscove's Modified Dulbecco's Medium (IMDM). Mouse bone marrow cells were collected and cultured in IMDM supplemented with murine MCSF (10 ng/ml) and concentrated lentivirus for a period of 72 h. Subsequently, the cells were cultured in fresh IMDM medium containing murine MCSF for an additional 48–96 h before being utilized for a phagocytosis assay.

### Protein complex expression and purification

We do not authenticate the cells used for protein expression. The wild-type human GPR84 was synthesized and cloned into pFastBac vector containing a bovine prolactin signal peptide followed by Flag-tag and $His_8$-tag at the N terminus. A fragment of engineered β2-adrenergic receptor N-terminal tail region (BN3) was fused GPR84 receptor at the N-terminal end to facilitate protein expression. To enhance the stability of the complex, the NanoBiT tethering strategy was used by fusing a LgBiT subunit at the C-terminus of the receptor[29]. The C-terminal residues G388-H396 was truncated and LgBiT was fused with a 15-amino acid linker (GSSGGGGSGGGGSSG). A dominant negative human $Gα_{i1}$ (DNG$α_{i1}$) containing four mutations (S47N, G203A, E245A, A326S) was cloned into the pFastBac vector[72]. Human $Gβ_1$ was fused with an N-terminal $His_6$-tag and a C-terminal HiBiT subunit connected with a 15-amino acid linker, was cloned into pFastBac dual vector together with human $Gγ_2$.

The expression and purification of scFv16 were achieved as previously described[73]. In brief, the scFv16 was expressed in Tni cells (Expression Systems, 94–002 F) and purified by nickel affinity chromatography before the C-terminal $His_8$-tag was removed by TEV protease. The protein was further purified by size exclusion chromatography using a Superdex 200 Increase 100/300 GL column (GE Healthcare). The monomeric peak fractions were pooled, concentrated and stored at −80 °C until use.

GPR84, DNG$α_{i1}$ and $Gβ_1γ_2$ were co-expressed in Sf9 insect cells (Expression Systems, 94-001 F) using Bac-to-Bac baculovirus expression system. Cells were infected with three types of viruses prepared above at the ratio of 1:1:1. After infection for 48 h at 27 °C, cell pellets were harvested and stored at −80 °C until use. Cell pellets were thawed in lysis buffer containing 20 mM HEPES, pH7.5, 50 mM NaCl, 10 mM MgCl2, 5 mM CaCl2, 2.5 μg/ml leupeptin, 300 μg/ml benzamidine. To facilitate complex formation, 10 μM 6-OAU, 25 mU/ml Apyrase (NEB), and 100 μM TCEP was added and incubated at room temperature for 2 h. The cell membranes were isolated by centrifugation at 30,700 g for 30 min and then resuspended in solubilization buffer containing 20 mM HEPES, pH7.5, 100 mM NaCl, 0.5% (w/v) lauryl maltose neopentylglycol (LMNG, Anatrace), 0.1% (w/v) cholesteryl hemisuccinate (CHS, Anatrace), 10% (v/v) glycerol, 10 mM MgCl2, 5 mM CaCl2, 12.5 mU/ml Apyrase, 10 μM 6-OAU, 2.5 μg/ml leupeptin, 300 μg/ml benzamidine, 100 μM TECP for 2 h at 4 °C. Insoluble material was removed by centrifugation at 38,900 g for 45 min, and the supernatant was incubated with Ni resin at 4 °C for 2 h. The resin was washed with a buffer A containing 20 mM HEPES, pH 7.5, 100 mM NaCl, 0.05% (w/v) LMNG, 0.01% (w/v) CHS, 20 mM imidazole, and 10 μM 6-OAU, 2.5 μg/ml leupeptin, 300 μg/ml benzamidine, 100 μM TECP. The complex was eluted with buffer A containing 400 mM imidazole. The eluate was supplemented with 2 mM CaCl2 and incubated with an anti-Flag M1 antibody resin overnight at 4 °C. Complex loaded on the Flag column was washed with 10 column volumes of buffer A supplemented 2 mM CaCl2. Then the complex was eluted by 3.5 column volumes of buffer A containing 5 mM EDTA and 200 μg/ml FLAG peptide. The complex was collected and concentrated using 100 kDa molecular weight cutoff concentrators (Millipore). Purified scFv16 was mixed with eluate at a 1.3:1 molar ratio. The sample was then loaded onto a Superdex 200 Increase 10/300 column (GE Healthcare) pre-equilibrated with buffer containing 20 mM HEPES pH 7.5, 100 mM NaCl, 0.00075% (w/v) LMNG, 0.00025% (w/v) GDN,

0.00015% (w/v) CHS, 10 μM 6-OAU and 100 μM TECP. Peak fractions of the complex were pooled and concentrated to 20 mg/ml for cryo-EM studies.

## Cryo-EM sample preparation and data acquisition

For cryo-EM grid preparation of the 6-OAU-GPR84-Gi complex, 3 μl of the purified complex at 20 mg/ml was applied onto a glow-discharged holey carbon grid (Quantifoil, Au200 R1.2/1.3). Grid was plunge-frozen in liquid ethane using Vitrobot Mark IV (Thermo Fischer Scientific). Cryo-EM imaging was performed on a Titan Krios electron microscope at 300 kV accelerating voltage using a Gatan K3 Summit direct electron detector with an energy filter. Micrographs were collected with a nominal magnification of ×81,000 using the EPU software in super-resolution mode with a calibrated pixel size of 1.07 Å and a defocus range of −1.2 to −2.2 μm. Each stack was acquired with an exposure time of 3.5 s and dose-fractionated to 32 frames with a total dose of 55 e·Å$^{-2}$. A total of 5307 movies were collected for 6-OAU-GPR84-Gi complex.

## Data processing, 3D reconstruction and modeling building

Image stacks were subjected to beam-induced motion correction using MotionCor2[74]. Contrast transfer function (CTF) parameters were estimated from motion-corrected images using Gctf[75]. Total of 8,056,512 particles of 6-OAU-GPR84-Gi complex were auto-picked using RELION 3.1[76] and then subjected to reference-free 2D classification to discard poorly defined particles. After several rounds of 3D classification, one well-defined subset with 628,450 particles was selected. Further 3D classification focusing the alignment on the receptor and complex, produced one high-quality subset accounting for 62,864 particles. These particles were subsequently subjected to 3D refinement, CTF refinement, and Bayesian polishing, which generated a map with an indicated global resolution of 3.0 Å at a Fourier shell correlation (FSC) of 0.143.

The Alphafold predicted structure of GPR84 was used as an initial model for model rebuilding and refinement against the electron microscopy map. The model was docked into the electron microscopy density map using Chimera[77] followed by iterative manual adjustment and rebuilding in COOT[78]. Real space refinement and rosetta refinement were performed using Phenix programs[79]. The model statistics was validated using MolProbity[80]. Structural figures were prepared in Chimera and PyMOL (https://pymol.org/2/). The final refinement statistics are provided in Supplementary Table 1. The extent of any model overfitting during refinement was measured by refining the final model against one of the half-maps and by comparing the resulting map versus model FSC curves with the two half-maps and the full model. Surface coloring of the density map was performed using UCSF Chimera[77].

## Molecular dynamics simulation and molecular docking

The Gi protein from the 6-OAU-GPR84-Gi cryo-EM structure obtained in this study was removed. The 6-OAU-GPR84 complex was subjected to molecular dynamics (MD) simulations in apo and holo states in a protein-lipid-water-ions environment. 6-OAU was removed from the structure and embelin, capric acid and 2-hydroxyl capric acid were docked to the hydrophobic cavity in absence of 6-OAU to obtain the respective docking poses. These poses were also employed as starting points for embelin, capric acid and 2-hydroxyl capric acid holo MD simulations. Lipid membrane was modeled in a 1:1 molar ratio of DOPC:POPC. CHARMM-GUI was used to assemble the simulation systems[81]. In the holo simulations, D66 was modeled in its protonated state, as reported for A-type GPCRs[82]; while in the apo simulations, D66 was modeled in its deprotonated state together with a sodium ion. The missing ICL3 (-100 missing residues) was modeled as a 16-residue loop made by joining the first 8 and last 8 residues of the missing ICL3 and constructed using Modeller by building 10,000 models. DOPE score

was used to choose the best model[83]. Besides 6-OAU, the presence of a cholesterol (CLR) molecule that binds to TM2-4 in the cryo-EM structure was also taken into account in MD simulations (with and without CLR). The CLR-binding site in the cryo-EM structure is similar to the CLR-binding site seen in β2-adrenergic receptor, which is proposed to allosterically modulate ligand binding at the orthosteric site[84]. Nonetheless, in our simulations, we find that CLR unbinds from GPR84 and remains unbound; Moreover, the presence of CLR has no notable influence on the ligand and protein dynamics in the apo and holo simulations, respectively.

For the native 6-OAU ligand, four replicas were simulated for each of the following four systems: apo-CLR, apo-noCLR, holo-CLR, and holo-noCLR. Each replica was simulated for ca. 1.25 μs, in total ca. 20 μs of simulations across all 4 states. For each of embelin, capric acid, and 2-hydroxyl capric acid, we also ran five replicas of MD simulations, with each replica simulated for 1 μs, in total 5 μs for the holo states for each ligand. The simulation systems comprised ca. 75,000 atoms. CHARMM36[85] forcefield was employed for the MD simulations. The systems were first subjected to an energy minimization for 10,000 steps and followed by gradual heating from 0 to 310 K for 500 ps, using a Langevin thermostat with heavy atoms restrained at 10 kcal mol$^{-1}$Å$^{-2}$ in an NVT ensemble. The heated systems were subjected to eight successive rounds of 1 ns equilibration steps. During the equilibration, protein and ligand-heavy atoms were subjected to harmonic restraints, and lipids were subjected to planar restraints to maintain bilayer planarity. The harmonic restraints for each step were relaxed progressively going from 10 to 0.1 kcal mol$^{-1}$Å$^{-2}$. The equilibrations were performed at a 1 fs timestep at $T = 310$ K and $P = 1$ bar using the Langevin thermostat and Nosé–Hoover Langevin barostat in NPT ensemble (Supplementary Fig. 12). The production runs were performed with a hydrogen mass repartitioning scheme with a timestep of 4.0 fs with a nonbonded cutoff at 12 Å[86]. Long-range electrostatics were evaluated with the Particle Mesh Ewald (PME) method. Protein and lipid bond lengths were constrained with the SHAKE algorithm. NAMD 2.14 was used for MD simulations[87] (see Supplementary Table. 3 for further MD simulation details). For the center-of-mass (COM) distance analysis of the holo simulations, the active site residues R172, T167, W360, Y69, N104 and F335 were considered to estimate native cavity COM. Distances between native cavity COM and the ligand COMs were calculated for the 5 μs MD trajectories for each holo state. Ligand RMSD was also calculated for the 5 μs MD trajectories for each holo state, using the respective crystal structure or docking pose as the reference. Receptors and ligands were prepared in Schrodinger environment, and Glide XP protocol with enhanced sampling and OPLS3 forcefield was used to perform the molecular docking[88]. The search space for docking was set with inner and outer box sizes of 10 and 26 Å, and the centroid of native ligand (6-OAU) was used as box center. Protein backbone RMSD for the apo and holo MD simulations shown in Supplementary Figs. 13 and 14.

## GTPγS binding assay

Studies on the potency of 6-OAU to activate GPR84 and how this was altered by variation at specific residues were conducted using a series of GPR84-Gα$_{i2}$ fusion protein[50,89]. Point mutation of residues predicted from the structural data to modify binding and or function of 6-OAU was introduced into such fusion proteins and expressed either stably in Flp-In T-REx 293 cells (Invitrogen, catalog number R78007) or transiently into HEK293T cells (ATCC, catalog number CRL-3216). The ability of varying concentrations of 6-OAU to promote binding of [$^{35}$S] GTPγS was then assessed as in our previous studies[51]. Briefly, membrane fractions of Flp-In T-REx 293 or HEK293T cells were incubated in buffer containing 20 mM Hepes pH 7.5, 5 mM MgCl$_2$, 160 mM NaCl, 0.05% fatty acid-free bovine serum albumin, and various concentrations of ligands. Then, [$^{35}$S] GTPγS (50 nCi per reaction) with 1 μM GDP was added and the mixture was incubated at 30 °C for 1 h. The reaction

was terminated by adding cold PBS buffer and the membrane fractions were collected by rapid vacuum filtration through GF/C glass fiber prefilters using a UniFilter FilterMate Harvester (PerkinElmer). After three additional washes with cold PBS, the filters were dried and incubated with MicroScint-20 (PerkinElmer). [³⁵S] GTPγS binding to $G_i$ was quantified by liquid scintillation spectroscopy. The data were analyzed by GraphPad Prism 6 (GraphPad Software).

## Radioligand binding assays

Compound 38 (9-(2-phenylethyl)–2-(2-pyrazin-2-yloxyethoxy)–6,7-dihydropyrimido[6,1-a]isoquinolin-4-one)[28] was tritiated and employed in specific binding assays to detect expression levels of wild-type GPR84 and various mutants as described previously[50,89], where it was designated [³H]G9543. Compound 140 (3-((5,6-diphenyl-1, 2,4-triazin-3-yl)methyl)−1H-indole)[36,37] was also tritiated and used in equivalent binding studies[36]. In each case, specific binding of the radioligand at the concentration of 5-fold higher than its $K_d$ at wild-type GPR84 was measured.

## Reporting summary

Further information on research design is available in the Nature Portfolio Reporting Summary linked to this article.

## Data availability

The data that support this study are available from the corresponding authors upon request. The 3D cryo-EM density map of 6-OAU-GPR84-$G_i$ has been deposited in the Electron Microscopy Data Bank under the accession numbers EMD-29645. Atomic coordinates for the atomic model have been deposited in the Protein Data Bank (PDB) under the accession numbers 8G05 [https://doi.org/10.2210/pdb8G05/pdb]. We used the following structures from the Protein Data Bank for our structural comparison analysis: BLT1 (PDBID 7VKT), S1PR1 (PDB ID 7TD3), LPAR1 (PDB ID 7TD0), EP2 (PDB ID 7CX2), CB2 (PDB ID 6PT0). The initial and final structures of GPR84 in the MD simulations studies are available in Figshare: https://figshare.com/s/e338e99709380e9a0aa1. The source data underlying Fig. 1, 3b, V, and Supplementary Fig. 3a are provided as a Source Data file. Source data are provided with this paper.

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

## Acknowledgements

We thank the Cryo-EM Center at University of Science and Technology of China for the support of cryo-EM data collection. This work was supported by the grant R35GM128641 from the National Institutes of Health (NIH) in the USA (to C.Z.), the Ministry of Science and Technology of China grant 2019YFA0904100 and the Natural Science Foundation of China grant T2221005 (both to W.G.). Work in the Milligan lab was supported by UK Research and Innovation Biotechnology and Biosciences Research Council (grant reference BB/T000562/1). H.F. and S.S. were supported by funding from the Biomedical Research Council of A*STAR in Singapore. The MD simulation work for this article was performed on resources of the National Supercomputing Centre, Singapore (https://www.nscc.sg).

## Author contributions

C.Z., W.G., and X.Z. conceived the project and designed the research with H.F., M.F., and G.M. Y.W. purified the protein for the cryo-EM study, X.Z. prepared and screened cryo-EM grids, collected cryo-EM data, and processed the data under the supervision of W.G. and C.Z. S.S. designed and performed computational docking and simulations studies under the supervision of H.F. X.C., J.Z., J.D., and S.C. performed macrophage functional assays under the supervision of M.F. L.J. and S.M. performed mutagenesis and pharmacology studies under the supervision of G.M. X.L., and G.L. assisted in protein production. C.Z. wrote the manuscript with the help from X.Z., W.G., S.S., H.F., M.F., and G.M.

## Competing interests

The authors declare no competing interests.
