## [Peer Review File · Nature Communications]

Pro-phagocytic function and structural basis of GPR84 signalingReviewers' Comments:

Reviewer #1:

Remarks to the Author:

The manuscript "Pro-phagocytic function and structural basis of GPR84 signaling" by Zhang et al, reports a cryo-EM structure of 6-OAU-bound human GPR84-Gi complex. Together with mutation data and MD simulation, the authors discussed the binding pocket of 6-OAU and receptor activation. GPR84 is a promising target for immune disease and cancer, this work provides molecular basis for the ligand discovery for GPR84. However, the manuscript organization and contents should be largely improved. There are several concerns which need to be addressed:

Major issues:

1. The phagocytosis assay presented in Figure 1 is not convincing to claim the pro-phagocytic effects is induced by GPR84-Gi signaling. More additional assays or controls are needed. GLPG1205 was reported as a NAM of GPR84, does it bind to the similar orthosteric binding pocket with 6-OAU in GPR84?
2. "The structure was determined to a global resolution of 3.0-Å by cryo-EM", according to the local resolution estimation in Fig. S2f, the resolution for the extracellular region is quite low. The electron density for the ligand and surrounding residues, as well as ECL2, should be shown.
3. In left panel of Figure 2, the author did not show the electron density for 6-OAU in the overall structure, while a strong piece of density is shown in the cytoplasmic parts of TM2-4. How does this density fit with cholesterol?
4. The expression data for mutants in Fig. 3d is missing, should be provided.
5. Does the disulfide bond between N-terminus and ECL2 affect the 6-OAU binding or the activation of the receptor?
6. MD simulations should be performed to support the docking poses of embelin, capric acid, and 2-hydroxy capric acid.
7. There are no mutagenesis data to verify the key residues that may related to GPR84 activation, such as Y3326.48, N1043.36, G1173.49.
8. How does K126 mutation affect the Gi coupling with GPR84?

Reviewer #2:

Remarks to the Author:

GPR84 is a proinflammatory receptor whose roles in lipid metabolism and immune regulation are emerging. In this manuscript, Zhang et al. demonstrated that the increase of macrophage phagocytosis depends on GPR84 and Gi. They solved a high-resolution cryoEM structure of GPR84 with the agonist 6-OAU. The structural model explains the ligand specificity and shows the unique ligand-binding features and structural motifs of GPR84. The authors further used computational tools for docking the native ligands and running MD simulations to identify the possible path as the entry or exit for the agonist. Overall, the paper is well-written and clear on the data and interpretation. The density map and the structural models are sound. The work would lead to further studies on the molecular mechanism of GPR84, which was poorly understood and may help therapy development. However, I still have some issues that need to be addressed by the authors before publishing the manuscript.

Major

1. Fig.1b, the error bars seem missing.
2. Fig. 3d, the mutation H352A has huge efficacy and potency change, but the authors did not explain the rationale for mutating this position or show the location of this residue. Why is this residue so crucial for the activation of GPR84?
3. The authors did not describe the docking process in the method section. Please indicate the parameters that the authors used in Glide/Schrodinger package.

4. The authors pointed out the unique features of GPR84-Gi interaction but did not show the comparison with the other GPCR-Gi pairs. The authors should support the claims like "in most of other Gi-coupled GPCR structures, the position 34.51 in the ICL2 is usually a hydrophobic residue." by providing some examples for the readers to compare. The authors compare the ligand binding pockets with the other lipid GPCRs in Figure 4. Receptors BLT1, S1PR1, and LPAR1 were solved with Gi-associated. Please provide a comparison of the Gi-binding mode of these related GPCRs either in the figures or in the supplementary data to support the claims.

Minor:

1. Fig. 1 legend the last sentence: "Each data point represents SD of data...". "Representing SD" cannot be accurate. Please correct the figure legend.
2. Fig. 4. For general readers, please describe how the receptor figures are generated. For example: "In each column, the cartoon model and surface representation are used for the same receptor."
3. The method section of cryoEM data processing indicates three subsets were used for 62,824 particles. Supplementary Fig. 2 suggested only one class after 3D classification was used. Please double-check and make them consistent.
4. The MD simulation should indicate the ligands' entry and exit routes. Do the authors only consider those paths as exit routes? Is there a reason for that?
5. GRY is a unique motif distinguishing the GPR84 from the other Class A GPCRs. Is it possible to introduce mutation on the Glycine or amino acids interacting with Glycine (like F55) and test the mutant in the function assay (like GTP γ S incorporation assay) to show that the GRY motif also plays the same critical role as the DRY motif?

We thank all reviewers for their constructive comments. Please see our detailed responses to the comments below. The reviewers' comments are in **blue** font and our responses are in **black** font.

REVIEWER COMMENTS

Reviewer #1 (Remarks to the Author):

The manuscript “Pro-phagocytic function and structural basis of GPR84 signaling” by Zhang et al, reports a cryo-EM structure of 6-OAU-bound human GPR84-Gi complex. Together with mutation data and MD simulation, the authors discussed the binding pocket of 6-OAU and receptor activation. GPR84 is a promising target for immune disease and cancer, this work provides molecular basis for the ligand discovery for GPR84. However, the manuscript organization and contents should be largely improved. There are several concerns which need to be addressed:

Major issues:

1. The phagocytosis assay presented in Figure 1 is not convincing to claim the pro-phagocytic effects is induced by GPR84-Gi signaling. More additional assays or controls are needed. GLPG1205 was reported as a NAM of GPR84, does it bind to the similar orthosteric binding pocket with 6-OAU in GPR84?

GLPG1205 may not bind to the same orthosteric binding pocket as that of 6-OAU in GPR84. Nonetheless, it functions as a specific antagonist of GPR84 and effectively inhibits GPR84 signaling induced by 6-OAU, even in the presence of 6-OAU binding (doi:10.1021/acs.jmedchem.0c00272). The fact that GLPG1205 abolishes 6-OAU-induced phagocytosis strongly suggests that this effect is mediated by GPR84. To further support this claim, we conducted CRISPR-Cas9-mediated knockout of GPR84 expression in primary macrophages and showed that the pro-phagocytic effect of 6-OAU was lost. These new findings are now presented in **Fig. 1c**.

2. “The structure was determined to a global resolution of 3.0-Å by cryo-EM”, according to the local resolution estimation in Fig. S2f, the resolution for the extracellular region is quite low. The electron density for the ligand and surrounding residues, as well as ECL2, should be shown.

We have included the suggested electron density map in the revised **Supplementary Figure 2g**. The density of the ligand and surrounding residues is clear for modeling.

3. In left panel of Figure 2, the author did not show the electron density for 6-OAU in the overall structure, while a strong piece of density is shown in the cytoplasmic parts of TM2-4. How does this density fit with cholesterol?

The binding pocket of 6-OAU is completely embedded within the 7TM domain of GPR84. Therefore, it is difficult to see the density of 6-OAU in the overall structure showing the cryo-EM density of the receptor. Nevertheless, we show the density of 6-OAU alone in **Figure 2**. The

density near the cytoplasmic region of TM2-4 fits cholesterol well. We have included such density in the revised **Supplementary Figure 2g**.

4. The expression data for mutants in Fig. 3d is missing, should be provided.

We thank the reviewer for the suggestion. We have now provided expression data for the mutants as measured by the specific binding of both a radiolabelled orthosteric 1,2,4 triazine antagonist ($[^3\text{H}]140$) and a similarly radiolabelled allosteric antagonist ($[^3\text{H}]38$) that is closely related to the clinically trialled GPR84 blocker GLPG1205. The data is shown in **Supplementary Data Figure 3a**.

In some cases, the mutants were unable to bind either of these ligands with high affinity as predicted for $[^3\text{H}]140$ from previous mutagenesis and homology modelling studies (Mahindra et al *J. Med. Chem.* 2022, doi.org/10.1021/acs.jmedchem.2c00804). However, for the Phe335Ala mutant for example, although it is unable to bind the orthosteric antagonist, which was also previously shown by Mahindra et al, its expression as a structurally intact protein was confirmed because this mutant does bind $[^3\text{H}]38$ effectively. For a further set of mutants, for example the Gly117 to either Ala or Asp, both radioligands bound with high affinity.

5. Does the disulfide bond between N-terminus and ECL2 affect the 6-OAU binding or the activation of the receptor?

It appears that it does. Alteration of Cys11 to Ala (which will prevent the formation of such a disulfide bond) completely eliminated G protein activation by 6-OAU. However, a caveat to this conclusion is that we did not observe binding of either the allosteric antagonist $[^3\text{H}]38$ or the orthosteric antagonist $[^3\text{H}]140$ in membranes of cells transfected to express GPR84 Cys11Ala and therefore disruption of this disulfide bridge likely limits correct folding of the receptor. We have included the G protein activation data in revised **Figure 3b** and the radioligand binding data in **Supplementary Data Figure 3a**. This frequently is the outcome when the conserved ECL2-TM3 disulfide bond in class A GPCRs is disrupted in various receptors.

It's worth noted that Liu et al (*Nature Communications*. 2023 Jun 6;14(1):3271) showed that they recorded reduced but still detectable cell surface expression of a C11A mutant, although they didn't probe its folding. Like us, they noted no agonist-induced signalling for this mutant.

6. MD simulations should be performed to support the docking poses of embelin, capric acid, and 2-hydroxy capric acid.

We thank the reviewer for the suggestion. We now include new MD simulations studies on embelin, capric acid, and 2-hydroxy capric acid starting from the docking poses in our revised manuscript. The data is presented in **Extended Data Figure 7 and 8**. Five copies (1us each) were run for each ligand, which totaled around 5 μs dynamics for each of embelin, capric acid and 2-hydroxy capric acid. We found that the four ligands primarily occupy the native binding pocket, consistent with the docking poses, with occasional forays to metastable sites such as S1, S2 and S3.

7. There are no mutagenesis data to verify the key residues that may related to GPR84 activation, such as Y3326.48, N1043.36, G1173.49.

We thank the reviewer for the suggestion. We have now included such mutagenesis data in our revised manuscript. The data is shown in **Figure 3b**. As noted in our manuscript, alteration of G117 to D eliminated function of 6-OAU whereas alteration of G117 to A didn't significantly affect 6-OAU signaling (**Fig. 3b**). For both G117A and G117D specific binding of both the allosteric antagonist [³H]38 and the orthosteric antagonist [³H]140 confirmed successful expression of these mutants (**Fig. S3a**).

Although residue 6.48 is most frequently W in class A GPCRs, in GPR84 this is replaced by Y322. Alteration of Y322 to W again resulted in reduced potency of 6-OAU whilst alteration to Ala all but eliminated responsiveness (**Fig. 3b**).

We also explored the contribution of N104 by converting this residue to Ala. The N104A mutant did not respond to any concentration of 6-OAU that we were able to test and, as such, this Asn indeed is likely to play a key role (**Fig. 3b**). Once more, however, the caveat to this conclusion is that we were unable to detect specific binding of [³H]38 or [³H]140 in membranes from cells transiently transfected with this mutant (**Fig. S3a**).

8. How does K126 mutation affect the G_i coupling with GPR84?

We state in our manuscript that '34.51 in the ICL2 is usually a hydrophobic residue', but 'In GPR84, this position is K126. As a result, there is no direct interactions between ICL2 of GPR84 and G_{ai}'. When we mutated K126 to Ala, this did not alter the potency of 6-OAU for G protein activation (**Fig. 3b**) or affect receptor expression (**Fig S3a**).

Liu et al (Nature Communications. 2023 Jun 6;14(1):3271) also reported that mutations at this position had no effect on G protein activation.

Reviewer #2 (Remarks to the Author):

GPR84 is a proinflammatory receptor whose roles in lipid metabolism and immune regulation are emerging. In this manuscript, Zhang et al. demonstrated that the increase of macrophage phagocytosis depends on GPR84 and G_i. They solved a high-resolution cryoEM structure of GPR84 with the agonist 6-OAU. The structural model explains the ligand specificity and shows the unique ligand-binding features and structural motifs of GPR84. The authors further used computational tools for docking the native ligands and running MD simulations to identify the possible path as the entry or exit for the agonist. Overall, the paper is well-written and clear on the data and interpretation. The density map and the structural models are sound. The work would lead to further studies on the molecular mechanism of GPR84, which was poorly understood and may help therapy development. However, I still have some issues that need to be addressed by the authors before publishing the manuscript.

Major

1. Fig.1b, the error bars seem missing.

We have revised our figure accordingly to include the error bars.

2. Fig. 3d, the mutation H352A has huge efficacy and potency change, but the authors did not explain the rationale for mutating this position or show the location of this residue. Why is this residue so crucial for the activation of GPR84?

Indeed, the relative efficacy of 6-OAU at H352A was low compared to wild type and potency was also substantially reduced. We initially expected that mutations of this residue wouldn't disrupt 6-OAU function since it doesn't interact with the ligand (**Fig. S3b**). However, it turned out not to be the case. This is probably explained by the poor expression of the mutant, as defined by low specific binding of [³H]38 and [³H]140 to H352A GPR84 (**Fig. S3a**). It is likely that the H352A mutant fails to fold correctly. We added a few sentences at the end of the second paragraph of section "Structure of the 6-OAU-GPR84-G_i complex and an occluded ligand binding pocket " to describe such findings. The position of this residue is shown in **Supplemental Data Figure S3b**.

3. The authors did not describe the docking process in the method section. Please indicate the parameters that the authors used in Glide/Schrodinger package.

We thank the reviewer for the suggestion. We now include details of docking in the Methods section.

4. The authors pointed out the unique features of GPR84-G_i interaction but did not show the comparison with the other GPCR-G_i pairs. The authors should support the claims like "in most of other Gi-coupled GPCR structures, the position 34.51 in the ICL2 is usually a hydrophobic residue." by providing some examples for the readers to compare. The authors compare the ligand binding pockets with the other lipid GPCRs in Figure 4. Receptors BLT1, S1PR1, and LPAR1 were solved with Gi-associated. Please provide a comparison of the Gi-binding mode of these related GPCRs either in the figures or in the supplementary data to support the claims.

We thank the reviewer for the suggestion. We now include a figure panel showing the comparison of the Gi-coupling to GPR84, BLT1, S1PR1, and LPAR1 at ICL2 in **Supplemental Data Figure 10**.

Minor:

1. Fig. 1 legend the last sentence: "Each data point represents SD of data...". "Representing SD" cannot be accurate. Please correct the figure legend.

The figure legend has been revised.

2. Fig. 4. For general readers, please describe how the receptor figures are generated. For example: "In each column, the cartoon model and surface representation are used for the same receptor."

We thank the reviewer for the suggestion. We have included such information in the figure legend stating that "In each column, the cartoon model and surface representation are used for the same receptor. The structures of the five receptors are placed side by side in each column after structural alignment, providing a consistent viewpoint from the same angle."

3. The method section of cryoEM data processing indicates three subsets were used for 62,824 particles. Supplementary Fig. 2 suggested only one class after 3D classification was used. Please double-check and make them consistent.

We confirm that only one class after 3D classification was used for obtaining the final map. We have corrected this error in the Method section.

4. The MD simulation should indicate the ligands' entry and exit routes. Do the authors only consider those paths as exit routes? Is there a reason for that?

The ligand dynamics from the large-scale MD simulations were analyzed to identify states where ligands deviated from the bound-sites observed in the cryo-EM structure or the docking structures. Our analysis suggests the existence of three potential exit routes, which were determined based on the proximity of ligands to membrane lipids or the extracellular matrix. However, due to the slow dynamics involved in the opening-up of the occluded pocket, it was challenging to observe a clear unbinding event within the timeframe of several microseconds of MD simulations. To capture a genuine binding event, simulations with several orders of magnitude longer periods would be required, demanding significant time and resources. Furthermore, extensive functional studies would be necessary for validation. Consequently, we believe that conducting such studies exceeds the scope of the present paper.

5. GRY is a unique motif distinguishing the GPR84 from the other Class A GPCRs. Is it possible to introduce mutation on the Glycine or amino acids interacting with Glycine (like F55) and test the mutant in the function assay (like GTP γ S incorporation assay) to show that the GRY motif also plays the same critical role as the DRY motif?

We thank the reviewer for the suggestion. As in our response to Reviewer 1, we have now included data on the G117D and G117A GPR84 mutants in our revised manuscript. The data is shown in **Figure 3b**. As noted in our manuscript, alteration of G117 to D, which would clash with F55, eliminated function of 6-OAU, whereas alteration of G117 to A didn't significantly affect 6-OAU signaling (**Fig. 3b**). For both G117A and G117D specific binding of both the allosteric antagonist [3 H]38 and the orthosteric antagonist [3 H]140 confirmed successful expression of these mutants (**Fig. S3a**).

Reviewers' Comments:

Reviewer #1:

Remarks to the Author:

The authors have addressed all my questions and concerns.

Reviewer #2:

Remarks to the Author:

The authors have done a great job responding to my initial comments. Especially additional mutagenesis assays confirming the contribution of the GRY motif.

However, I have one issue with the revised manuscript:

In the new Fig.1 panel c, the legends say "GPR84KD". Does that mean "GPR84 knockdown"? In the manuscript, the author indicates GPR84 was knockout. Please provide evidence of the GPR84 knockout or knockdown.

We thank all reviewers for their constructive comments. Please see our detailed responses to the comments below. The reviewers' comments are in **blue** font and our responses are in **black** font.

Reviewer #1 (Remarks to the Author):

The authors have addressed all my questions and concerns.

Thanks.

Reviewer #2 (Remarks to the Author):

The authors have done a great job responding to my initial comments. Especially additional mutagenesis assays confirming the contribution of the GRY motif.

However, I have one issue with the revised manuscript:

In the new Fig.1 panel c, the legends say "GPR84KD". Does that mean "GPR84 knockdown"? In the manuscript, the author indicates GPR84 was knockout. Please provide evidence of the GPR84 knockout or knockdown.

It is GPR84 knockdown. We have provided evidence for GPR84 knockdown in Supplementary Figure 1c.